# Clinical Management of Low Anterior Resection Syndrome: Review of the Current Diagnosis and Treatment

**DOI:** 10.3390/cancers15205011

**Published:** 2023-10-16

**Authors:** Ruijia Zhang, Wenqin Luo, Yulin Qiu, Fan Chen, Dakui Luo, Yufei Yang, Weijing He, Qingguo Li, Xinxiang Li

**Affiliations:** 1Department of Colorectal Surgery, Fudan University Shanghai Cancer Center, Shanghai 200032, China; 17301050248@fudan.edu.cn (R.Z.); 21111230037@m.fudan.edu.cn (W.L.); 19301050224@fudan.edu.cn (Y.Q.); 19301050220@fudan.edu.cn (F.C.); dkluo17@fudan.edu.cn (D.L.); 20111230052@fudan.edu.cn (Y.Y.); wjhe15@fudan.edu.cn (W.H.); 2Department of Oncology, Shanghai Medical College, Fudan University, Shanghai 200032, China

**Keywords:** rectal cancer, low anterior resection syndrome, diagnosis, assessment

## Abstract

**Simple Summary:**

Low anterior resection syndrome (LARS) significantly impacts the lives of 80% of patients who undergo sphincter-preserving surgery, often leading to diminished quality of life and social avoidance. However, a lack of systematic standards and varied measurement and treatment approaches have hindered optimal management and understanding of LARS. This study seeks to synthesize findings from up until 2023 and establish unified diagnostic criteria and management protocols to facilitate seamless integration of findings from specifical clinical trials into clinical application. The aim is to enable comparable research outcomes and enhance clinical methodologies, potentially offering a coherent framework that could shape future research directions and improve clinical outcomes in LARS management. We proposed that in clinical treatment, the severity of LARS should be assessed using at least one symptom assessment questionnaire and one scale related to quality of life. In clinical trials, a new criterion including the LARS score, EORTC QLQC30, and anorectal manometry is suggested to be adopted by following the research. A stepwise classification model was set up for the standardized clinical management of LARS.

**Abstract:**

Background: Low anterior resection syndrome (LARS) is a series of bowel dysfunction symptoms, including altered bowel frequency, irregular bowel rhythms, fecal incontinence, and constipation. LARS occurs in 80% of patients undergoing sphincter-preserving surgery, affecting patients’ quality of life along with social avoidance. Different measurements and treatments have been raised to deal with LARS, but no systematic standard has been developed. Objective and Methods: To promote the standardization of clinical trials and clinical management of LARS, this review summarizes the latest findings up until 2023 regarding the diagnostic criteria, assessment protocols, and treatment modalities for postoperative LARS in rectal cancer. Results: The diagnostic criteria for LARS need to be updated to the definition proposed by the LARS International Collaborative Group, replacing the current application of the LARS score. In both clinical trials and clinical treatment, the severity of LARS should be assessed using at least one symptom assessment questionnaire, the LARS score or MSKCC BFI, and at least one scale related to quality of life. Anorectal manometry, fecoflowmetry, endoscopic ultrasonography, and pelvic floor muscle strength testing are recommended to be adopted only in clinical trials. After analysis of the latest literature on LARS treatment, a stepwise classification model is established for the standardized clinical management of LARS. Patients with minor LARS can start with first-line treatment, including management of self-behavior with an emphasis on diet modification and medication. Lamosetron, colesevelam hydrochloride, and loperamide are common antidiarrheal agents. Second-line management indicates multi-mode pelvic floor rehabilitation and transanal irrigation. Patients with major LARS should select single or several treatments in second-line management. Refractory LARS can choose antegrade enema, neuromodulation, or colostomy. Conclusions: In clinical trials of LARS treatment between 2020 and 2022, the eligibility criteria and evaluation system have been variable. Therefore, it is urgent to create a standard for the diagnosis, assessment, and treatment of LARS. Failure to set placebos and differentiate subgroups are limitations of many current LARS studies. Randomized controlled trials comparing diverse therapies and long-term outcomes are absent, as well. Moreover, a new scale needs to be developed to incorporate the patient’s perspective and facilitate outpatient follow-up. Though the establishment of a stepwise classification model for LARS treatment here is indispensable, the refinement of the guidelines may be improved by more standardized studies.

## 1. Introduction

Colorectal cancer is the third most common cancer and ranks second in mortality worldwide; in one-third of cases, tumors develop in the rectum [1]. The incidence and mortality of colorectal cancer in China have shown an increasing trend. According to the Chinese cancer statistics, 376,000 new cases and 191,000 deaths occurred in 2018, and the incidence and mortality rates of colorectal cancer in China ranked third and fifth among all malignant tumors [2].

Radical surgical resection is the preferred treatment method for rectal cancer. Sphincter-preserving surgery (SPS) is routinely recommended for middle and upper rectal cancer (cT2-4, N0-2, M0) [2]. The development of neoadjuvant therapy, as well as improvements in surgical technology, including the introduction of circular staplers, have increased the indications for surgery, and up to 80% of patients with rectal cancer are currently eligible for SPS [3,4]. However, radical resection of rectal cancer is associated with severe postoperative complications and a significant decrease in quality of life, and patients suffer a great physical, emotional, and financial burden [5]. Therefore, in addition to improving survival, attention should be devoted to patient survivorship.

Low anterior resection syndrome (LARS) refers to a series of bowel dysfunction symptoms that may occur after low or ultra-low anastomosis for rectal cancer [6,7]. These symptoms, which include altered bowel frequency, irregular bowel rhythms, fecal incontinence, and constipation, have a negative impact on patients’ quality of life [8], leading to anxiety, depression, social impairment, and social avoidance [9]. The literature suggests that 80% of patients undergoing SPS will develop LARS [10]. Among these patients, 25% have minor LARS [11], whereas 35–60% of patients have major LARS lasting for several years [12,13]. The major symptoms of LARS are fecal incontinence (65.2%) and clustering (66.7%) [14].

The pathophysiologic mechanism underlying the development of LARS includes reduction in rectal reservoir capacity and compliance, and the interrupted continuity of the enteric nervous system [15]. Risk factors for LARS include temporary stomata, anastomotic height, and neoadjuvant and adjuvant radiotherapy, whereas the type of anastomosis and the choice of surgical approach do not affect the incidence or severity of LARS. Recent research efforts have focused on intraoperative neuromonitoring, pelvic floor rehabilitation before stoma closure, and transanal irrigation as prevention strategies to reduce symptoms [16]. Studies suggest that diffusion-weighted imaging (DWI) and microRNA analysis may reduce the use of neoadjuvant therapy through individualized treatment to maintain anorectal function [17].

Although in most patients, symptoms decrease within 12 months postoperatively, a significant proportion of patients have long-term problems such as defecation difficulties 15 years after surgery. Generally, the symptoms of LARS decrease with time [18] and the key indicators of quality of life improve yearly in patients who undergo ileostomy retraction [19]. However, LARS is a long-term progressive condition rather than a transient postoperative change [20]. Therefore, LARS requires long-term standardized care, and approximately 50% of patients with major LARS show improvement in response to standardized interventions [21]. This review summarizes the latest findings in the literature regarding the diagnostic criteria, assessment protocols, and treatment modalities for postoperative LARS in rectal cancer. The updated information on LARS research provided in this review will be useful for the establishment of management guidelines for clinical treatment and suggestions for the development of clinical trials related to the treatment of LARS. In conducting this review, we systematically searched for and extracted articles from various renowned databases, including PubMed and EMBASE, concentrating on publications from up to 2023 concerning diagnostic criteria, assessment protocols, and treatment modalities for postoperative LARS in patients with rectal cancer.

## 2. Diagnosis Systems for LARS

### 2.1. Pathophysiology and Risk Factors for LARS

LARS is defined as bowel dysfunction after rectal resection, leading to a detriment in quality of life, as reported in *The Lancet Oncology* in 2012 [4]. Pelvic surgery, particularly LAR for rectal cancer, may damage the internal anal sphincter (IAS) and associated sympathetic nerves, contributing to LAR syndrome [22] (Table 1). Studies have shown correlations between postoperative IAS pressures, the length of the remaining rectum, and incontinence, indicating potential parasympathetic nerve or surgical damage affecting IAS function [23,24]. Reduced anal canal sensation, possibly due to nerve damage during LAR, impacts the ability to distinguish between flatus and feces [25]. Some studies have demonstrated significant differences in sensitivity thresholds and anal canal sensitivity postoperation, linking decreased sensation to incontinence [26]. The disappearance of the rectoanal inhibitory reflex (RAIR) plays a critical role in differentiating between flatus and feces and has been studied extensively in preoperative and postoperative patients showing symptoms of LAR syndrome [27]. The loss of RAIR, influenced by the length of the remaining rectum, seems pivotal to function post-LAR, affecting defecatory function and increasing incontinence [28]. After total mesorectal excision (TME), the remaining rectum contributes less to capacity and compliance due to its reduced reservoir function. Studies indicate a significant reduction in rectal capacity and compliance postoperation, affecting the urge and maximum tolerable volume for defecation [29]. Radiation therapy in conjunction with TME impacts the functional outcomes, contributing to higher defecation frequency and reduced compliance [30].

As for the risk factors, LARS exhibits a substantial correlation with the height of the anastomosis. Some studies reported that possessing a remaining rectum ≥4 cm in length was correlated with markedly enhanced functional outcomes (such as the rectoanal inhibitory reflex and rectal capacity) compared to those with less than 4 cm of residual rectum [31]. Additionally, radiotherapy holds a pivotal role in the multifaceted treatment approach to rectal cancer. While neoadjuvant radiotherapy has notably enhanced sphincter preservation—through effective downstaging and downsizing—and has fortified local tumor control, its adverse influence on postoperative functional outcomes and quality of life has been recurrently documented [32]. Following anastomotic leakage in patients with rectal cancer, irrespective of whether a secondary fistula operation is undertaken, local inflammation induced by the leakage can provoke infection or fibrosis around the anastomotic stoma. This results in diminished rectal volume and compliance, impacts bowel function, elevates LARS scores, and reduces quality of life [33,34]. A defunctioning stoma is employed in low and ultra-low anastomoses for anterior resection of rectal cancer, whereas transverse prophylactic colostomy is more commonly used in patients undergoing neoadjuvant therapy. These procedures can effectively mitigate the incidence and severity of anastomotic leakage by diverting the fecal matter, minimizing mechanical irritation to the anastomosis, and lowering the infection around the anastomosis. Currently, the correlation between defunctioning stomata and LARS is still debated; some research implies that a defunctioning stoma can enhance postoperative anal function in patients, subsequently improving their quality of life [35].

### 2.2. Diagnostic Criteria for LARS

LARS is divided into two categories according to specific clinical manifestations. The first category is mainly manifested as fecal incontinence and increased frequency of bowel movements, which is known as urgent incontinence, in which patients with major LARS may defecate more than 10 times/day [36]. The second category is mainly manifested as constipation, incomplete emptying, and defecation difficulty, and it is known as evacuation disorder. Patients may have only one of these clinical conditions, or alternatively experience both types [37]. This definition is too broad to clearly identify the diagnostic criteria because the symptoms of LARS are highly variable. Therefore, in clinical practice and in academic studies, the LARS score is used instead of the previous definition to diagnose LARS and estimate the severity of the disease [38].

Although the LARS score has been validated in different regions [38,39,40], it remains merely a scale for the rapid screening and assessment of LARS patients instead of a complement to the definition. Therefore, the scale may underestimate the effects of LARS on evacuatory dysfunction and overestimate its impact on quality of life in a large portion of patients [8].

In 2020, the LARS International Collaborative Group published a consensus that distinguished between symptoms and outcomes among diverse clinical presentations of LARS (Table 2 and Table 3) and redefined LARS as “patients who have at least one of these symptoms and result in at least one of these consequences after low anterior rectal resection” [8], which can only be diagnosed when a causal relationship between symptoms and outcomes exists [8]. The consensus was informed by 325 patients, specialized surgeons, and healthcare professionals, incorporating multiple stakeholders and prioritizing patient views. Since the consensus was published, this definition has been used by several LARS clinical management guidelines [41].

### 2.3. Assessment and Testing of LARS

The assessment of the severity of LARS usually includes grading scales, anorectal manometry, endoscopic rectal ultrasound, and fecoflowmetry (Table 4) [14].

Several scoring systems can be used to assess bowel function after SPS, including the Wexner Incontinence Grading Scale [42], the Vaizey Scale [43], the William Scale [44], the Fecal Incontinence Severity Index [45], the Memorial Sloan Kettering Cancer Centre Bowel Function Index (MSKCC BFI) [46], the LARS score [47], and the European Organization for Research and Treatment of Cancer (EORTC) QLQC30/QLQCR29 [48]. Currently, only two scale systems, MSKCC BFI and the LARS score, are designed specifically for LARS [49]. The other scales are mostly used for the assessment of LARS-related symptoms or quality of life. The MSKCC BFI, an 18-item validated scoring instrument, is mostly used for research purposes. This tool has limited clinical value, consisting of a time-consuming questionnaire that lacks weight for different symptoms. The LARS score was developed as a screening tool for identifying LARS [47]. It is a 5-item questionnaire that can stratify LARS patients into “no LARS”, “minor LARS”, and “major LARS”. It can be used for outpatient screening, late adverse reaction assessment, and long-term monitoring, but its capability to detect changes over time has been questioned as sensitivity to change has not yet been formally tested [41].

Although the MSKCC BFI and the LARS score have similar values [50], the LARS score is considered a better instrument for screening intestinal dysfunction because of its convenience. However, a high LARS score does not indicate severe subjective feelings of intestinal dysfunction [51]. The score does not include indicators for individual quality of life. Additionally, the test does not consider the significant differences in age and gender [52]. Because fecal incontinence is the most important symptom affecting quality of life in patients with LARS, some studies also use the Wexner Incontinence Grading Scale and other scoring systems to evaluate LARS [53,54]. SPS can also cause symptoms in other organs, such as sexual dysfunction and urinary dysfunction. Therefore, a relevant quality of life scale may also be used [55].

Anorectal manometry can objectively evaluate anal sphincter function and rectal volume [56]. It is a widely used scale for the evaluation of LARS [9]. Anorectal manometry can measure rectal anal resting pressure, maximum contraction pressure, rectal anal inhibitory reflex, rectal volume, and the compliance data of patients. This method can be used during treatment and follow-up, and it is useful for clinical trials because of its ability for continuous monitoring [41,57]. Although the results of preoperative anorectal manometry are relevant to the outcomes of multivariate analysis, there is no reliable evidence indicating that the results can predict postoperative LARS [58]. Fecoflowmetry can be used to evaluate postoperative anorectal motor function. The defecated volume, flow time, mean and maximum flow rates, and time to maximum flow rate are obtained by simulating diarrhea to calculate the defecation flow curve [59]. The efficacy of fecoflowmetry for evaluating postoperative anorectal motility is equivalent to that of the symptom severity score and anorectal manometry [60]. Endoscopic ultrasonography can detect lacerations of the sphincter and changes in the puborectal angle in patients with LARS. However, the sample size of the relevant trial is too small for the routine use of endoscopic ultrasonography [61].

Next, we conducted a search in the PubMed database for terms related to LARS and clinical trials, limiting the time frame to 2020–2022. We summarized a total of 15 clinical trials that adopted LARS assessment methods (Figure 1). The results showed that 83% of the studies used symptom assessment tools such as the LARS score. 13% of the studies used quality of life assessment tools such as EORTC QLQC30, while only 7% of the studies used objective evaluation indicators such as anorectal manometry and pelvic floor muscle strength testing. This demonstrated that the methods used in current clinical trials to assess postoperative LARS symptoms and their severity are not standardized, often leading to difficulties in similar study comparisons and rendering the summarization of clinical trial results to guide future management and treatment of LARS patients bleak.

## 3. Research Progress on LARS Treatment

The main limitations of current clinical trials of LARS are the small sample size, lack of consistent inclusion and exclusion criteria, and inconsistent evaluation methods. Conducting a high-quality meta-analysis is thus difficult. A standardized treatment for LARS remains to be established in the absence of multicenter clinical trials with large samples. Most studies and treatment strategies address specific subgroups of LARS patients and mainly consider the impact of urgent incontinence on quality of life. Current treatments include self-behavior management, dietary modification, pharmacotherapy, transanal irrigation, pelvic floor rehabilitation, neuromodulation, antegrade enema, and temporal/permanent colostomy.

### 3.1. Self-Behavior Management and Dietary Adjustment

Self-behavior management is the most simple intervention measure and includes increasing exercise, supplementing water, dietary adjustments, and correct defecation [14]. Although there is little evidence that such measures are effective in controlling symptoms, they do not have side effects or complications and help patients manage their symptoms [41,62,63]. Dietary adjustment includes avoiding foods that may soften feces, such as caffeine, citrus, spicy foods, and alcohol, and increasing dietary fiber. The first-line treatment of LARS also includes fiber supplements, although different types of dietary cellulose perform differently. Adding psyllium can significantly decrease the frequency of fecal incontinence, whereas carboxymethylcellulose (CMC) increases the frequency of fecal incontinence [64]. A prospective study of nearly 60,000 older women found that long-term dietary fiber intake reduces the risk of fecal incontinence, particularly liquid incontinence [65].

### 3.2. Antidiarrheal Drugs

These drugs belong to the first-line treatment of LARS, and the most commonly used drugs are loperamide, ramosetron, atropine, and raceanisodamine, which inhibit the abnormally high frequency of intestinal movement to treat LARS with urgency and incontinence. The clinical data of these drugs are mostly derived from the treatment of fecal incontinence in other gastrointestinal diseases such as IBS. Studies show that the use of probiotics does not alter the postoperative bowel function associated with low anterior resection syndrome [66].

Loperamide is a μ-opioid receptor agonist and the most commonly used prescription drug for LARS [67]. It is most effective in attenuating defecation frequency and defecation incontinence symptoms. These drugs can effectively improve defecation frequency and slow intestinal contractions and peristalsis [68]. However, in patients with LARS associated with emptying disorders, these drugs should be used with caution because they can aggravate constipation [69]. Loperamide can also treat perianal pain or itching with a sitz bath or local ointment [41].

5-HT3 antagonists suppress bowel movement after SPS, especially spastic hyperkinesia. Their efficacy in the treatment of IBS defecation incontinence and their safety were demonstrated in clinical trials. Lamosetron can reduce the population of patients with major LARS. After 4 weeks of treatment (5 μg/d), the Wexner Incontinence Grading Scale, defecation urgency, and daily defecation frequency significantly decreased, and only 7% of patients reported side effects such as constipation or defecation difficulties [70]. However, these studies mostly recruited male patients [71], and the safety and feasibility of lamosetron therapy for female patients needs to be further verified. Bile acid chelator is a particularly important treatment for patients receiving neoadjuvant radiotherapy because pelvic radiotherapy can lead to the poor absorption of bile acid. However, the type of bile acid chelator affects the efficacy of the treatment. More than 80% patients show no response to colestyramine treatment. However, in a study including 30 valid cases, colesevelam hydrochloride improved the absorption of bile acid and decreased diarrhea and the urgency and frequency of defecation in cancer patients. More than 60% of patients choose to take long-term medication [72].

### 3.3. Enteroclysis

If LARS symptoms are still present after conservative treatment, transanal irrigation (TAI) can effectively improve intestinal function and quality of life. The principle of TAI is to wash the colon, thereby delaying rectal emptying. TAI can last for more than 1 day or can be used to improve incomplete emptying or planned defecation [41]. TAI improves compliance, satisfaction, bowel function, and quality of life in patients with fecal incontinence and constipation [73,74,75]. A single-center randomized controlled trial of 23 patients showed that TAI was more effective than percutaneous tibial nerve stimulation (PTNS) [76]. Although the enema process requires training, certain psychological acceptance, and the guidance of experienced medical staff [77], studies show that patients generally believe that this is an acceptable treatment. It also demonstrates that self-control of defecation is very important for patients. In addition, the safety of TAI is an important factor. Intestinal perforation is a rare complication, with an incidence of 2–6 cases/million [78].

The specific frequency and the amount of fluid of enteroclysis are not clearly defined and depend on the patient’s preference. However, the overall effect is better with a greater amount of flushing fluid [79]. The starting time and treatment duration of TAI remain to be determined, and most studies choose 6 months of treatment. Some guidelines recommend TAI 30 days after surgery [80], whereas others recommend waiting for 6 months to evaluate [14]. Many patients who use TAI stop the treatment at 3–12 months, although the long-term benefits after the cessation of TAI treatment remain unclear. Martellucci et al. reported an increase in the LARS score and a decrease in quality of life at 3 months after the cessation of TAI [75]. This suggests that patients may require long-term TAI. Rosen et al. showed that at 12 months after TAI, the scores are similar between patients who undergo TAI for 3 months and those who undergo continuous TAI treatment [81]. The inconsistency in these results may be related to the bias caused by the small sample size, the benefit to patients with specific subtypes, and the different experimental designs. Large-scale studies in patients stratified according to LARS subtype are necessary.

In 2021, a study that included 44 patients showed that Xilei powder, metronidazole, and gentamicin combined with dexamethasone retention enteroclysis had good clinical efficacy; the underlying mechanism may be related to the stimulation of the rectal mucosa to release 5-HT substances and regulate defecation function [55]. This innovative therapy may lead to the development of new treatments for LARS and fecal incontinence.

In recent years, the concept of preventive TAI to protect rectal function after ileostomy closure has been proposed. This therapy is more effective for improving symptoms within 3 months after surgery than supportive care and medication alone, and early use of TAI does not have serious side effects [82]. Therefore, starting TAI in the early postoperative stage of LARS could be beneficial for patients at high risk. Independent high-risk factors include neoadjuvant radiotherapy and tumor distance from the anus [11].

Another enema-based method is antegrade continence enema, which can be performed using percutaneous endoscopic colostomy, appendectomy, or ileostomy, and it involves small amounts of fluid. A meta-analysis of 17 studies showed that 74% of patients had successful antegrade continence enema [83]. Didailler et al. reported a significant improvement in LARS scales in patients receiving standard antegrade continence enema, and 88% of the patients did not need colostomy in the future [84]. Approximately one-third of patients reported local pain, whereas chronic abdominal pain was rare. Complications include stoma stenosis in appendectomy or ileal neoadrenal stomata. Percutaneous endoscopic cecalostomy can protect the stoma. Because the maximum sample size of published studies was 60 [85], this method requires further verification. In addition, this procedure requires experienced medical teams and equipment and it is difficult to achieve the best results.

### 3.4. Pelvic Floor Rehabilitation Training

Pelvic floor rehabilitation (PFR) includes pelvic floor muscle training (PFMT), biofeedback training (BFT), and rectal balloon training (RBT). It improves bowel function in patients who have failed in first-line management by restoring muscle strength and the coordination of pelvic contractions, as well as lowering the patient’s threshold for perceived rectal distension. The effectiveness of PFR in the treatment of LARS has been demonstrated, and several studies have concluded that multimodal pelvic floor rehabilitation (including at least one PFR treatment) can improve defecation frequency, incontinence attacks, and quality of life [86,87]. A randomized controlled trial combining all PFR modes and electrical stimulation therapy was conducted in 2021 and is still ongoing. Its aim is to address the methodological limitations of the above studies and to obtain more definitive evidence [88].

PFMT generally refers to Kegel and anal sphincter exercise training, which should be performed under the guidance of a professionally trained and dedicated nurse practitioner [89]. Kegel exercise strengthens the pelvic floor muscle groups by consciously and actively contracting the anal levator and anal sphincter muscles, which increases the firmness of the pelvic floor muscle groups and decreases anal drop. The exercise can promote the recovery of the damaged anal sphincter and nerves, improve rectal compliance, and thus control voluntary defecation [87]. This method is easy to learn, is not limited by time or place, and does not increase the patient’s pain. Although it is commonly used for the non-surgical treatment of fecal incontinence [90], its efficacy in patients with LARS is inconclusive. While some studies suggest that PFMT may improve LARS and restore anal function [91], other studies failed to show a significant improvement in fecal incontinence scores after PFMT [92]. However, studies are consistent in showing that patients can benefit from PFMT, especially patients with symptoms of defecation urgency. In 2018, Ding Yuzhen et al. proposed the use of Kegel exercises combined with hydrotherapy for LARS based on local inflammatory edema in the early postoperative period. Hydrotherapy can promote local blood circulation, reduce edema, and reduce anal swelling. The results showed that the 85 patients in the treatment group had better somatic function, social function, and emotional status than the control group at 6 months postoperative, suggesting that this treatment is effective [93].

Biofeedback (BF) can optimize the patient’s motor response using visual and auditory signals, lower the threshold of rectal dilatation sensation, and synchronize the voluntary contraction of the external anal sphincter to provide feedback on rectal dilatation [41]. BF combined with PFMT reduces the incidence of LARS [94]. BF alone also improves anorectal maximum resting pressure as well as rectal volume [92]. BF can accelerate the recovery of pelvic function and reduce the duration of major LARS during the period in which LARS symptoms are most severe. BF treatment is administered at a lower frequency than other therapies, often 1 to 2 times per week [95]. BF can be performed safely for a defined period of time and is considered one of the best treatment options for LARS [96]. The patient benefits are greater with early BF [97].

The mechanism of RBT includes progressive reduction of balloon volume to improve rectal sensitivity and help patients distinguish between different rectal filling volumes. Balloon training decreases rectal sensitivity and counteracts the recto-anal inhibitory reflex in conjunction with voluntary anal contractions [98]. RBT is often used in combination with PFMT, although the addition of RBT does not significantly improve patient outcomes [98], and its benefits are limited to the control of defecation urgency and lifestyle adaptations.

### 3.5. Electrical Nerve Stimulation

Electrical nerve stimulation includes noninvasive nerve stimulation, such as posterior tibial nerve stimulation and transcutaneous tibial nerve stimulation, and implanted nerve stimulation, which is known as sacral nerve stimulation.

Sacral nerve stimulation, which is also known as sacral neuromodulation (SNM), is a minimally invasive surgical intervention that is effective for different etiologies of FI [99]. In more than two-thirds of patients with fecal incontinence, permanent implants provide long-term benefits, and studies indicate that SNM is a safe and effective treatment for fecal incontinence [100]. However, a placebo effect plays a role [101]. Patients with a history of radiation therapy respond poorly to SNM, and the efficacy of SNM depends on the anastomosis site [102]. SNM is divided into two phases—the testing phase, which is often referred to as peripheral nerve assessment or transcutaneous nerve assessment, and permanent implantation, and both of these can be performed on an outpatient basis. Patients who show improvement in bowel incontinence during the testing phase can undergo permanent implantation, which is usually located in the gluteal region and can be administered with a small handheld device [103]. Because LARS patients require more frequent changes in treatment regimens than the average patient with fecal incontinence, adequate follow-up is essential to ensure the long-term effectiveness of the treatment [104]. The most common serious adverse event associated with SNM is implant site infection, and approximately 3% of patients who receive permanent implants require external placement of the device [105]. Although SNM is more costly than conservative treatment and biofeedback, studies have shown the cost-effectiveness of SNM-integrated quality-adjusted life years [100].

PTNS was first proposed in 2015 by Troncoso et al. at the European Society of Colorectal Diseases [62]. This method is less invasive, simpler, and cheaper than SNM but its efficacy is inferior to that of SNM and similar to that of posterior tibial nerve stimulation [41]. PTNS is another minimally invasive procedure in which a small electrode is inserted above the medial ankle adjacent to the posterior tibial nerve and a surface electrode is bonded below the arch of the foot; the two electrodes are connected to a stimulator for 30 min. Patients who received PTNS showed better long-term outcomes than the sham treatment group without nerve stimulation [106]. PTNS has better short-term efficacy than drug therapy, whereas its efficacy is comparable with that of drug therapy after 1 year [107].

## 4. Recommended Treatment Guidelines for LARS

Based on the above latest findings and guidance from previous relevant guidelines and consensus, this review proposes a stepwise classification model for the standardized clinical management of LARS (Figure 2). Different treatment levels should be progressively adopted and patients should be treated symptomatically according to the type of LARS. However, there are many treatment modalities for LARS and a definitive comparison of the efficacy of each therapy is lacking. Therefore, on the basis of first-line treatment, patients should select the appropriate second-line treatment considering the personal economic level, local medical conditions, and the psychological and physical health status. Physicians should confirm the presence of LARS using the latest consensus definition of LARS [8] to exclude underlying lesions and specific etiologies of diarrhea, such as perianal lesions related to contamination or radiation. Patients are evaluated using the LARS scale or MSKCC BFI and at least one scale involving the patient’s quality of life, and if a patient presents with an urgent incontinence type of LARS, a scoring system such as the Wexner Incontinence Grading Scale and the Fecal Incontinence Severity Index can be used to supplement the assessment of characteristic symptoms. Assessment using anorectal manometry, for example, is not necessary in the absence of special studies or follow-up needs. This review uses the LARS scale to distinguish between patients with minor and major LARS. Patients with minor LARS can take progressive steps to control their symptoms according to a stepwise treatment plan, which aims to treat LARS using simple, cost-effective, and noninvasive methods. Patients with major LARS and those at high risk of developing major LARS postoperatively should begin first-line therapy as soon as possible in the early postoperative period and initiate one or more second-line therapies as soon as conditions permit to protect rectal function and reduce the duration of major LARS.

First-line treatment includes management of self-behavior with emphasis on diet modification and medication. In addition to increasing exercise, hydration, and proper bowel movements, a professional dietitian can be consulted to correct poor eating habits. Counseling can be sought when mental and emotional health, social and daily activities, and intimate relationships are affected, and the use of leakproof clothing is a common protective measure for patients with LARS [41]. An appropriate diet should include high fiber and low-fat foods, whereas alcohol, cold drinks, and spicy foods should be avoided. Patients with the urgent incontinence type may use psyllium to increase dietary fiber intake and soluble fiber supplements are recommended for patients with voiding disorders to assist in reducing constipation and improving stool consistency. However, high levels of insoluble fiber may aggravate diarrhea, the frequency of bowel movements, and bloating [108]. Most of the drugs currently used for LARS are aimed at patients with acute incontinence LARS, and symptom typing of LARS is important for their adequate application.

In cases in which first-line therapies are not effective or in those with persistent rectal dysfunction for more than 1 month, one or more second-line therapies may be considered depending on individual circumstances. More than half of the patients, especially those with minor LARS, show good symptom control if second-line treatment is administered according to the criteria [101]. If the above treatments are ineffective or if major LARS has been present for more than 1 year, treatment of recalcitrant LARS should be undertaken [67], which can include antegrade continence enema and sacral nerve stimulation. With standard treatment, if major LARS persists 2 years after surgery, a stoma is recommended [62]. However, there is no evidence showing that stomata improve quality of life in patients with recalcitrant LARS. Patients receiving a stoma must be fully informed of the pros and cons of this surgery, including the advantages (no urgency, no incontinence, and no anal pain) and disadvantages (parastomal hernia, prolapse, dermatitis, and leakage) of the stoma [67].

## 5. Discussion

The findings of recent studies and previous reviews indicate that the diagnostic criteria for LARS need to be updated to the definition proposed by the LARS International Collaborative Group. Patients should be screened using the LARS score or MSKCC BFI and at least one scale assessing quality of life. The Wexner Incontinence Grading Scale and Fecal Incontinence Severity Index are used for the assessment of acute incontinence LARS, and tests such as anorectal manometry, fecal flow rate assessment, endoscopic ultrasonography, and pelvic floor muscle strength testing are available for follow-up and research. The first-line treatment strategy consists of dietary modification, self-behavior management, and psychological counseling. Patients with prolonged disease lasting >1 month can be treated with one or more second-line treatment programs, and patients with severe disease or duration of disease >1 year can be considered for a continuous LARS treatment program. The treatment of LARS requires long-term patient self-management, including behavior modification, dietary modification, pelvic floor rehabilitation, sacral nerve stimulation, and transanal irrigation. The efficacy of commonly used treatments is also dependent on patient implementation and requires an emphasis on patient education and follow-up [10].

The diagnosis and treatment of LARS have been investigated extensively, and clinical trials have reported good efficacy and assessment results, providing a variety of options for the treatment of LARS. However, the specific curative effects of these treatments and the reliability of the results remain unclear, and the inclusion criteria, assessment systems, and analysis methods differ between studies, making meta-analyses difficult and hampering the development of standardized treatments. Therefore, randomized controlled trials comparing therapies such as anal irrigation, biofeedback therapy, sacral nerve stimulation, and combinations of these therapies are important. At the same time, failure to set placebos, small sample sizes, and failure to differentiate between subgroups are the limitations of many studies. The inclusion of a placebo or sham control can reduce the symptoms of fecal incontinence by more than 30% [101], which exceeds the improvement obtained with some treatments. Large-sample multicenter randomized controlled trials need to be performed in the future, and great attention should be paid to patient inclusion criteria and unified assessment systems and analysis methods. Changes in patient age or clinical status, preoperative radiotherapy, and anastomotic location all have a potential role in postoperative anal function recovery and treatment response and should be analyzed in studies with a 2-year follow-up. Although LARS is treated as a collection of symptoms, in clinical practice, the symptoms vary greatly from patient to patient, and several studies have shown that different treatments may be appropriate for different subgroups of LARS patients [98], underscoring the need for patient staging in future research.

The LARS definition consensus proposed by the LARS International Collaborative Group adopted a tripartite perspective including physicians, patients, and caregivers, although the consensus has not been widely used and accepted because of its late publication. This model is the first to incorporate the patient’s perspective into a study related to the diagnosis and treatment of LARS, reflecting the importance of decision-sharing between clinicians and patients in the diagnosis and treatment of LARS [41]. LARS patients are highly individualized and the appropriate subgroups vary according to the assessment system and treatment modalities. Therefore, the risks and alternatives associated with any procedure or operation should be discussed with the patient and individually tailored to their condition and status. Although there are currently many scale systems for LARS assessment, the LARS scale or the MSKCC BFI and at least one scale involving the patient’s quality of life are recommended in this review for the comprehensive assessment of patients. However, because the LARS score is only a screening tool, it does not include follow-up of dynamic changes in the patient’s condition. In addition, it does not include assessment of other processes, such as the evaluation of urinary status and sexual dysfunction, and a new scale needs to be developed. Future questionnaires should incorporate the patient’s perspective and include the evaluation of quality of life to avoid the use of multiple questionnaires and facilitate outpatient follow-up. Evaluation systems also need to incorporate the new definition proposed by the LARS International Collaborative Group Consensus to complement the parts of the definition that are not covered.

## 6. Conclusions

Overall, this review indicated a need to refine the diagnostic criteria for low anterior resection syndrome (LARS) to align with proposals from the LARS International Collaborative Group, emphasizing comprehensive patient assessment, involving LARS scores, quality of life scales, and other physiological tests. A multifaceted approach to treatment, involving dietary, behavioral modifications, and psychological counseling is crucial, especially for long-term management. Diverse treatment options from extensive clinical trials are available, yet their specific curative impacts and reliability are still unclear due to varying study methodologies and a lack of standardized treatments. Therefore, well-structured, large-scale, randomized controlled trials are imperative to develop standardized treatments and improve patient-specific interventions, incorporating patient perspectives and quality of life evaluations for more holistic, patient-centered care, facilitating better postoperative outcomes, and long-term management. Moreover, there’s a consensus on including patient and caregiver perspectives in LARS studies, underlining the need for decision-sharing in LARS diagnosis and treatment.

## Figures and Tables

**Figure 1 cancers-15-05011-f001:**
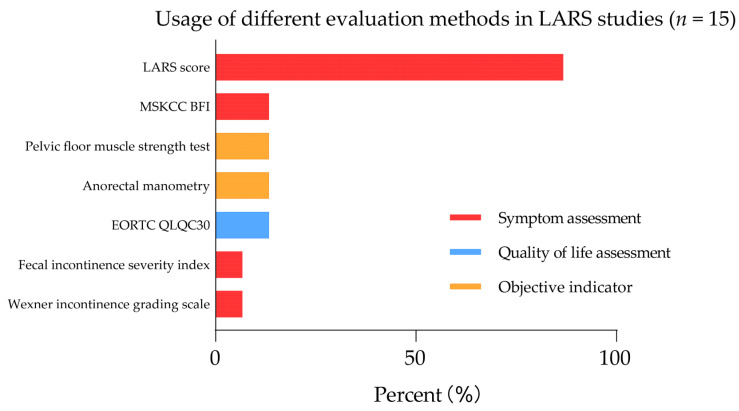
Statistical chart of assessment methods used in LARS clinical trials.

**Figure 2 cancers-15-05011-f002:**
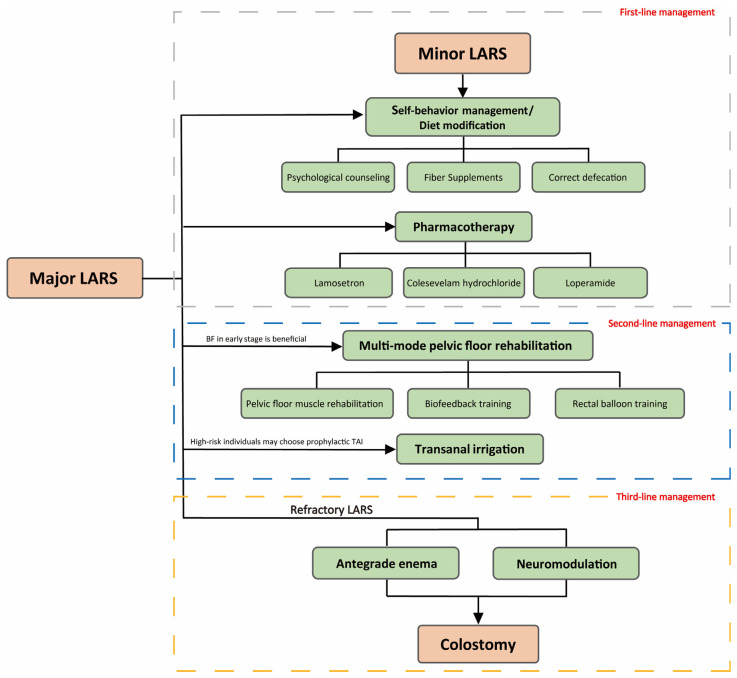
A flowchart shows the suggested clinical criteria for management of LARS.

**Table 1 cancers-15-05011-t001:** Pathophysiology and risk factors associated with LARS.

Pathophysiology Factors	Risk Factors
Internal anal sphincter dysfunction	Anastomosis location
Decrease in anal canal sensation	Neoadjuvant/adjuvant radio-chemotherapy
Interrupted continuity of the enteric nervous system	Anastomotic leakage
Reduction in rectal reservoir capacity and compliance	Defunction stoma

**Table 2 cancers-15-05011-t002:** Eight symptom complexes identified by the LARS International Collaborative Group.

Symptoms	Paraphrase
Variable, unpredictable bowel function	Unexpected diverse changes in bowel function, including unpredictable movements, etc.
Altered stool consistency	Compared to preoperative stool frequency
Increased stool frequency	Compared to preoperative stool frequency
Repeated painful stools	Pain on urge, on passing a bowel movement, and/or after passing a bowel movement
Emptying difficulties	Difficulty emptying the bowel for any reason; a feeling that the bowel has not completely emptied after passing a bowel movement; and the need to return to the toilet multiple times to empty the bowel
Urgency	The need to rush to the toilet to defecate and/or the inability to delay passing a bowel movement
Fecal incontinence	The unintended passage of a large volume of fecal material
Soiling	The involuntary passage of a small amount of material onto clothing or sanitary items

**Table 3 cancers-15-05011-t003:** Eight consequences identified by the LARS International Collaborative Group.

Consequences	Paraphrase
Toilet dependence	Psychologically or physically dependent on the toilet
Preoccupation with bowel function	Pay excessive attention to rectal function in daily life, have fear and/or anxiety about bowel control
Dissatisfaction with bowels	Dissatisfaction with bowel function despite knowing the possible postoperative changes in rectal function before the surgery
Strategies and compromises	Coping strategies need to be planned to manage bowel function and compromise on the events it affects
Mental and emotional wellbeing	Have a negative effect on the mental state and mood of the patient, affect the overall sense of wellbeing
Social and daily activities	Have a negative impact on social and daily activities
Relationships and intimacy	Have a negative impact on relationships and feelings of intimacy
Roles, commitments, and responsibilities	Affect the patients’ social roles, commitments, and family responsibilities they originally assumed

**Table 4 cancers-15-05011-t004:** Common evaluation methods of LARS.

Subjective Indicator		Objective Indicator
Symptom Assessment	Quality of Life Assessment	
LARS score	EORTC QLQC29	Anorectal manometry
MSKCC BFI	EORTC QLQC30	Fecoflowmetry
Wexner incontinence grading scale	EORTC QLQC38	Endoscopic ultrasonography
Fecal incontinence severity index	GIQLI	Pelvic floor muscle strength test

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
