# Peer review of "Clinical Management of Low Anterior Resection Syndrome: Review of the Current Diagnosis and Treatment"

_cancers, 2023, doi:10.3390/cancers15205011_

Round 1
Reviewer 1 Report
Thank you very much for giving me the opportunity to review this well structured review, which concerns a very relevant topic.
Overall, I enjoyed reading it, however, I have some comments and suggestions, which I believe will improve the paper.
Introduction
”Sphincter-preserving surgery (SPS) and anterior rectal resection are routinely recommended for middle and upper rectal cancer”
Anterior rectal resection is sphincter preserving as well. Please rephrase.
”improvements in surgical technology and the prevalence of circular staplers”
"Prevalence" may not be the best word to use in this context. I’ll suggest “improvements in surgical technology including the introduction of circular staplers”, or something like that...
”The literature suggests that 80% of patients undergoing SPS will develop LARS[10]. Among these patients, 20% have mild LARS[11], whereas 30–55% of patients have severe LARS lasting for several years[12, 13]”
I’ll suggest underlining that these proportions are across all patients undergoing SPS, because the prevalence depend on the risk factors you’re mentioning in the next section.
“Among these patients” refers to those who will develop LARS, hence the proportions should add up to 100. Please rephrase.
For consistency, consider using the terms “no, minor, major LARS” throughout the paper when you’re referring to results based on the LARS score.
”The mechanism underlying the development of LARS is related to the reduced volume of the rectum, poor compliance, and the interrupted continuity of the enteric nervous system accompanied by bowel rehabilitation [15]. ”
This sentence is unclear. What is “poor compliance” (compliance with what?) and how can “bowel rehabilitation” be an underlying mechanism?
1.2. Assessment and testing of LARS
”…and the European Organization for Research and Treatment of Cancer (EORTC) QLQC30/QLQCR38[34] ”
QLQCR38 must be replaced by QLQCR29
”It can be used for outpatient screening, late adverse reaction assessment, and long-term monitoring, but the scoring system is not suitable for evaluating the therapeutic effect[27].”
Please check the publication – it is not referenced correctly. The LARS score may not have formally been tested for sensitivity to change, still, several studies have been able to show a change over time.
3. Recommended treatment guidelines for LARS
”Patients must be informed prior to stoma surgery because at least 20% of post-SPS complications involve temporary stomas that cannot be retracted.”
In this context, where you’re mentioning a stoma as a treatment option for major LARS, discussing the proportion of reversal of a temporary stoma is not that relevant? (Maybe I’m just missing a point here).
Recovery of patients who have undergone a stoma is highly informative, including the advantages (no urgency, no incontinence, no anal pain) and disadvantages (parastomal hernia, prolapse, dermatitis, leakage) of the stoma[53].
Could you use another phrase than “highly informative”?
Figure 1
Please add a title to Figure 1. I understand that the boxes on the right side aim to show which treatment modalities belong to each “treatment level”. However, using the exact same type of black lines as the ones used to illustrate the “treatment-progression-flow” makes it a bit confusing. I’ll suggest dropping the black lines on the right hand side of the figure, and illustrate the three “treatment-levels” in a different graphical way in the figure.
A few suggestions added in "comments to the authors"
Reviewer 2 Report
The authors propose a comprehensive review of literature regarding LARS syndrome after low anterior resection for colorectal cancer. The article presents the current scoring systems and in differrent regions and focus upon the necessity of improvement of the standardisation for this pathology.
Some aspects could improve the manuscript before publication:
1. A paragraph stating the databases and time frame for the articles included in the review.
2. A separate paragraph with the Conclusions of the present work
Reviewer 3 Report
This manuscript by Ruijia Zhang et al. reviews current guidelines for diagnosis, assessment and treatment of low anterior resection syndrome (LARS). This manuscript is well written and is a comprehensive description of the evaluation and treatment of LARS. Authors have done a good job with tiered description of the treatment approach and inclusion of multidisciplinary assessments and intervention.
Despite this, there have been several reviews on the same topic published in the literature in recent years (Surgical Oncology Volume 43, August 2022, 101691; Cancers 2023, 15(3), 778, etc). and this review does not add much to the existing literature. Depending on the goals of the journal, this article could be published as it is a good summary. However, if the journal seeks to answer new questions, this article does not provide any new information. My only suggestion for authors is to discuss in detail about pathophysiology and risk factors associated with LARS, and measures to reduce the development of LARS.
Round 2
Reviewer 1 Report
Line 155: ”It can be used for outpatient screening, late adverse reaction assessment, and long-term monitoring, but the scoring system is not suitable for evaluating the therapeutic effect[27].”
This sentence should be re-phrased to reflect the cited paper, which says "The LARS score may be less useful as an outcome parameter in monitoring treatment effects, as its capability for detecting changes over time has been questioned."
My suggestion is
”It can be used for outpatient screening, late adverse reaction assessment, and long-term monitoring, but it's capability for detecting changes over time has been questioned as sensitivity to change has not yet been formally tested. [27]
